# The Addition of Biochar and Hyper-Thermal Inoculum Can Regulate the Fate of Heavy Metals Resistant Bacterial Communities during the Livestock Manure Composting

**Qingjun Zhang** [1,2,†]**, Tong Zhu** [1,*,†]**, Qingxiang Xiao** [1] **and Ning An** [1]

1   School of Mechanical Engineering and Automation, Northeastern University, Shenyang 110004, China; zhangqingjunsir@163.com (Q.Z.); x1712110903@163.com (Q.X.); an173024600396@163.com (N.A.)
2   Liaoning Urban Construction Design Institute Co., Ltd., Fushun 113008, China
*   Correspondence: tongzhu@mail.neu.edu.cn
†   These authors contributed equally to this work.

**Abstract:** In the present investigation the effects of biochar and hyper-thermal inoculum on the heavy-metal-resistant bacteria (HMRB) during livestock manure composting were studied. An experiment was performed on composting livestock manure and wheat straw amended with biochar and hyper-thermal inoculum. Physicochemical properties, enzyme activity, heavy metals (HMs), and bacterial activities were monitored, and a comprehensive assessment was analyzed during the composting process. The results showed that the dominant phyla of Proteobacteria, Bacteroidota, Actinobacteriota, and Chloroflexi were enriched, but this was not the case with Firmicutes. The abundance of Galbibacter, Thermobifida, Sphaerobacter, and Actinomadura was significantly different in CT15 and BHCT15. In addition, this study showed that the selected factors are less correlated with HMRB compared with the CT group. Therefore, this study could provide new insights into the effect of biochar and hyper-thermal inoculum amendments on the fate of HMRB under HMs and high temperature stress during livestock manure composting.

**Keywords:** heavy metal resistance; biochar; hyper-thermal inoculum; livestock manure; composting; bacterial activities

## 1. Introduction

Agricultural waste sharply increases every year with the rapid development of animal husbandry. The livestock manure contains organic contaminants and heavy metals (HMs), which may cause serious pollution to the environment if not properly treated [1]. In general, composting is the most efficient way to achieve organic matter recycling and alleviate environmental risks [2,3]. The bacterial bioactivity plays a momentous role in the bio-oxidation of composting, and the microorganisms exert different mechanisms to resist external stress factors such as HMs and high temperature (HT) [4,5]. Recently, the innovating hyper-thermal composting process with hyper-thermal inoculum amendments has attracted widespread attention, as it is superior in efficiency and quality compared with the conventional process [6]. Furthermore, biochar is considered an ideal additive for reducing the mobility and bioavailability of HMs. Therefore, more studies now investigate biochar and hyper-thermal inoculum amendments in the composting process [7].

Although many papers have extensively studied bacterial diversity during composting, knowledge regarding the effects of biochar and hyper-thermal inoculum addition on heavy-metal-resistant bacteria (HMRB) is limited [8]. The HT and concentration of HMs limit the expression and bioactivity of sensitive microbes, except HMRB, which can resist hyper-thermal environments and HM stress [9,10]. Many studies have reported that the addition of inorganic and organic additives can reduce potential environmental hazards to HMRB. Awasthi et al. [11] reported that an amendment of biochar mitigates HM mobility

and persistent toxicants to promote the metabolic activity of HMRB. Zhou et al. [12] showed that an application of biochar can provide a survival environment for HMRB and relevant enzymatic activities. Awasthi et al. [4] indicated that biochar-amended compost exhibited superior HMRB with driving physicochemical properties and toxic HMs. In addition, the combined amendment of biochar and bacterial inoculum can significantly regulate a functional bacteria consortium [13]. Meanwhile, hyper-thermal environment stress can change the structure of organic matter to mediate the passivation of HMs [14]. Liu et al. [15] reported that hyperthermophilic composting can accelerate the polycondensation reactions by intensifying the oxidation of humic acid to enhance quality and quantity. On the other hand, it has been reported that humic acid has superior metal binding capacities and can improve the abundance and diversity of HMRB [16,17]. Therefore, the application of hyper-thermal inoculum can influence HMRB by regulating the potential bacterial host. At present, there are still limitations to the abundance and diversity of HMRB under HM and HT pressure [18]. In particular, how biochar and hyper-thermal inoculum amendments affect HMRB succession and the correlations of HMRB with physicochemical parameters, enzyme activity, and HMs are unknown. Therefore, the specific performance of HMRB to HM and HT stress is worth exploring further.

Thus, the present study provides new insights into the effect of biochar and hyper-thermal inoculum amendments on the fate of HMRB under HM and HT stress during livestock manure composting. The objective of this study was to (1) investigate the impact of biochar and hyper-thermal inoculum on the relative abundance and diversity of HMRB, (2) demonstrate the predictive correlation between HMRB with physico-chemical parameters, enzyme activity, and HMs, and (3) explore the succession of an integrated bacterial consortium with a combined amendment of biochar and bacterial inoculum. Further, a novel and ecological hyperthermophilic composting approach with a biochar amendment was developed.

## 2. Materials and Methods

### 2.1. Compost Materials and Preparations

Livestock manure and wheat straw were obtained from local farmland in Heping District, Shenyang, China. The bamboo biochar and hyper-thermal inoculum were collected from Dongyuan Environmental Technology Co., Ltd., Shenyang, China, owned by the research team. The biochar was prepared from biomass feedstock via slow and dry pyrolysis at a temperature of 200–300 °C at atmospheric pressure for 4 h. Each pile consisted of a mixture of 35 kg of chicken manure, 85 kg of cow dung, and 20 kg of wheat straw. Afterwards, the raw material of each pile was homogenized for use as the main raw materials. Pile A was thoroughly mixed with 15 kg of saw dust as a control, while Pile B contained 7.5 kg of hyper-thermal inoculum, and Pile C contained 7.5 kg of hyper-thermal inoculum mixed with 7.5 kg of biochar, respectively indicated as CT, HCT, and BHCT. The end products of the hyper-thermal compost were used as the inoculum for the HTC and BHCT processes. The main physico-chemical properties of the raw materials, CT, HCT, and BHCT are shown in Table 1. The basic parameters of the biochar were the moisture (1.53 ± 0.08%), the pH (8.69 ± 0.05), and the C/N (132.25 ± 1.09), and the parameters of the hyper-thermal inoculum were also the moisture (10.32 ± 0.53%), the pH (8.15 ± 0.06), and the C/N (80.25 ± 1.32). The moisture content of the initial mixture for each pile was adjusted to nearly 55%. The compost was set into three 800 × 500 × 500 mm troughs. The continuous aeration rate was set to 0.2 L·kg$^{-1}$ dry matter·min$^{-1}$. The composting pile was turned up on Days 5 and 10 to ensure oxygen supply. An approximately 200 g sample was taken from four locations at a depth of 50–100 cm in three piles on Days 1, 3, 5, 7, 9, 11, 13, and 15. About 20 g of each homogenized sample was stored at −20 °C for DNA extraction, and other samples were stored at 4 °C for corresponding analysis.

**Table 1.** The main physicochemical properties of the raw materials, CT, HCT, and BHCT.

| Parameter | Composting Raw Materials | | | | | | Initial Mixture | | |
|---|---|---|---|---|---|---|---|---|---|
| | Chicken Manure | Cow Dung | Wheat Straw | Biochar | Saw Dust | Hyper-Thermal Inoculum | CT | HCT | BHCT |
| Weight (kg) | 105 ± 1.51 | 255 ± 1.63 | 60 ± 1.01 | 7.5 ± 0.23 | 22.5 ± 0.32 | 15.0 ± 0.23 | 153.21 ± 1.34 | 155.19 ± 1.69 | 156.6 ± 1.76 |
| Moisture (%) | 67.85 ± 0.80 | 65.21 ± 0.93 | 7.64 ± 0.12 | 1.53 ± 0.08 | 8.63 ± 0.09 | 10.32 ± 0.53 | 56.61 ± 0.29 | 52.07 ± 0.68 | 53.72 ± 0.76 |
| pH | 6.79 ± 0.07 | 6.83 ± 0.05 | 7.23 ± 0.03 | 8.69 ± 0.05 | 8.45 ± 0.03 | 8.15 ± 0.06 | 7.69 ± 0.09 | 8.02 ± 0.11 | 7.92 ± 0.06 |
| TOC (%) | 55.31 ± 1.03 | 51.81 ± 0.87 | 62.35 ± 0.86 | 78.03 ± 0.56 | 58.76 ± 0.56 | 63.78 ± 0.91 | 52.72 ± 0.87 | 53.48 ± 0.76 | 64.39 ± 0.92 |
| TN (%) | 2.13 ± 0.02 | 1.62 ± 0.06 | 5.32 ± 0.06 | 0.59 ± 0.05 | 0.49 ± 0.02 | 0.79 ± 0.04 | 1.74 ± 0.02 | 1.65 ± 0.07 | 1.80 ± 0.09 |
| C/N | 25.97 ± 1.02 | 31.98 ± 1.08 | 11.72 ± 0.96 | 132.25 ± 1.09 | 121.21 ± 1.06 | 80.25 ± 1.32 | 30.03 ± 0.76 | 32.41 ± 0.84 | 35.77 ± 0.93 |

Note: Values indicate mean ± standard deviation based on the samples with 3 replications. TOC: total organic carbon; TN: total nitrogen; C/N ratio: carbon-to-nitrogen ratio.

## 2.2. Physicochemical Analysis and Enzyme Activities

Thermometers were used for the daily measure of the temperature of the environment and the piles. The pH value and electrical conductivity (EC) were detected with a pH meter and a conductivity meter, respectively [19,20]. The $NO_3^-$-N and $NH_4^+$-N concentrations were analyzed by the spectrophotometric method [6]. Total nitrogen (TN) and total organic carbon (TOC) concentrations were measured with an Elementar analyzer to calculate the C/N ratio [21]. Moisture content (MC) was determined following the method of Esperón et al. [22]. Germination index (GI) was determined as described by Sun et al. [23]. Total element concentrations of P, Cr, Pb, Cd, As, and the available heavy metals were determined using a method described previously [24]. The activities of protease, phosphatase, cellulase, urease, and arylsulfatase (ARS) were measured according to previous research [25].

## 2.3. Bacterial Community Detection

The total DNA was extracted using the E.Z.N.A.® soil DNA Kit (Omega Bio-tek, Norcross, GA, USA), and the quality and concentration of the extracted DNA were determined with the NanoDrop ND-2000 spectrophotometer (Thermo Scientific, Wilmington, NC, USA). The DNA was then sequenced with the Illumina Miseq platform. The primers 338F (ACTCCTA CGGGAG GCAGCAG) and 806R (GGACTACHV GGGTWT CTAAT) were used to amplify the V3–V4 hypervariable regions of the 16S rRNA gene, using the thermocycler PCR system (GeneAmp 9700, ABI, CA, USA) [5,26].

## 2.4. Statistical Analyses

Annotations and evaluations of operational taxonomic units were performed by rank abundance. The distribution and relative abundance of HMRB were determined by visualizations and ternary diagrams. A paired Fisher's exact test was conducted to evaluate divergence analysis. Principal component analysis (PCA) was conducted for beta diversity analysis, and redundancy analysis (RDA) was used to plot the relationship between heavy-metal-resistant bacteria and selected factors. The bioinformatic analysis and RDA were performed using the online Majorbio Cloud Platform (www.majorbio.com accessed on 31 July 2021).

## 3. Results and Discussion

### 3.1. Species Assessment and Composition Analysis of Bacteria during Composting

#### 3.1.1. Annotation and Evaluation of Operational Taxonomic Units (OTUs)

The rank-abundance distribution curve of nine treatments at the OTU level are shown in Figure 1. In the rank-abundance distribution curve, each curve can be used to compare the evenness and richness of species by the width and smoothness. A wider curve reflects a richer species, and a flatter curve represents a higher evenness of species. As shown in Figure 1, the BHCT group was the widest and flattest in the transverse and longitudinal direction, which indicated that the BHCT group had more abundant and uniform bacterial communities than other treatments during different composting stages. It is suggested that

amendments of biochar and hyper-thermal inoculum can regulate the abundance of HMRB under HM and HT stress, enhancing bacterial reproduction [11,14].

## Rank-Abundance curves

**Figure 1.** Rank-abundance distribution curve at the OTU level.

3.1.2. Relative Abundance of HMRB during Composting

The relative abundance of HMRB from the phylum to species level during composting is shown in Figure 2. The ternary phase diagrams explain the specific weight relationship in the phyla of the bacterial community among CT, BHCT, and HCT during three different composting stages (Figure 3), which vividly reflects the influence of the biochar and hyper-thermal inoculum on the HMRB. The circle indicates the bacterial communities at the phylum level, and the size of the circle represents the bacterial community abundance. As shown in Figures 2 and 3, the dominant phyla were Firmicutes, Proteobacteria, Bacteroidota, Actinobacteriota, and Chloroflexi, consistent with the results of Awasthi et al. [27]. Among all phyla, Firmicutes abundance was 32.23%, 18.92%, and 55.58% in CT1, BHCT1, and HCT1 in the initial phase, respectively. Bacteroidota accounted for 18.54%, 24.41%, and 6.53%, and Bacteroidetes was identified with 7.39%, 12.30%, and 10.57%. In CT3, BHCT3, and HCT3, Firmicutes was observed with 87.86%, 60.94%, and 71.59%, Proteobacteria was observed with 4.48%, 8.93%, and 4.98%, and Actinobacteriota was identified with 4.79%, 12.53%, and 8.03%. Notably, in CT15, BHCT15, and HCT15, Chloroflexi was observed with 2.07%, 11.74%, and 2.13%. Additionally, obvious distinct compositions of heavy-metal-resistant bacteria were also identified at the class to species level Figure 2b–f. Among the class of CT3 and BHCT3, Bacilli affiliated with Firmicutes was observed with 83.41% and 55.90%, Bacteroidia accounted for 0.82% and 8.93%, Actinobacteria was identified with 4.36% and 12.29%, and Gammaproteobacteria abundance was 1.41% and 9.88%. Deinococci was observed with 2.31% and 2.95% in BHCT3 and HCT3. The order Bacillales had the lowest abundance, while Flavobacteriales were the most abundant in the thermophilic phase. Additionally, obvious distinct compositions of Thermomicrobiales were observed in BHCT15 and HCT15. Meanwhile, the families Weeksellaceae and Thermomicrobiaceae

had higher values in BHCT3 and BHCT15. Notably, genera *Muricauda*, *Moheibacter*, and *Sphaerobacter* were abundant in BHCT1, BHCT3, and BHCT15, respectively. Species of uncultured *Moheibacter* and unclassified *Sphaerobacter* were more dominant in BHCT3 and BHCT15.

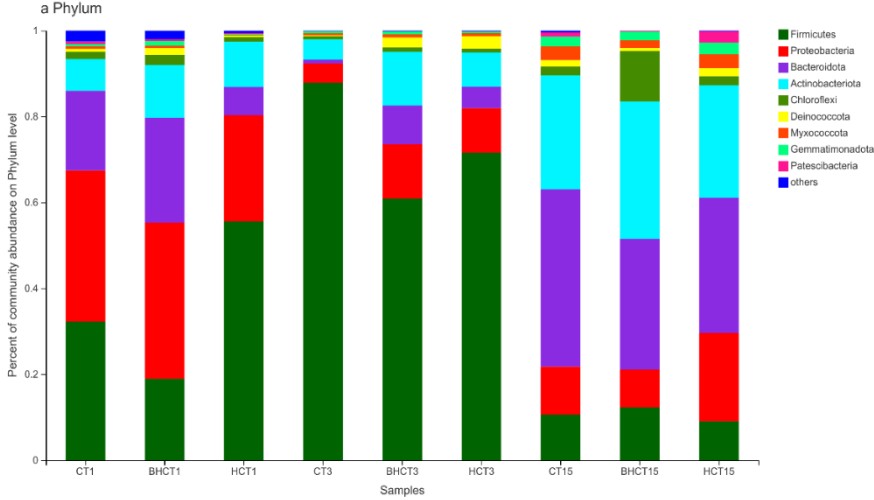

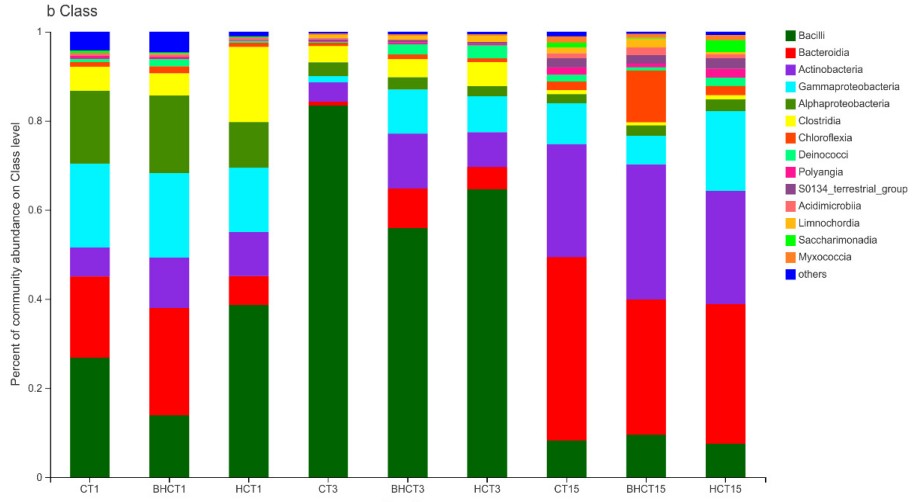

**Figure 2.** *Cont.*

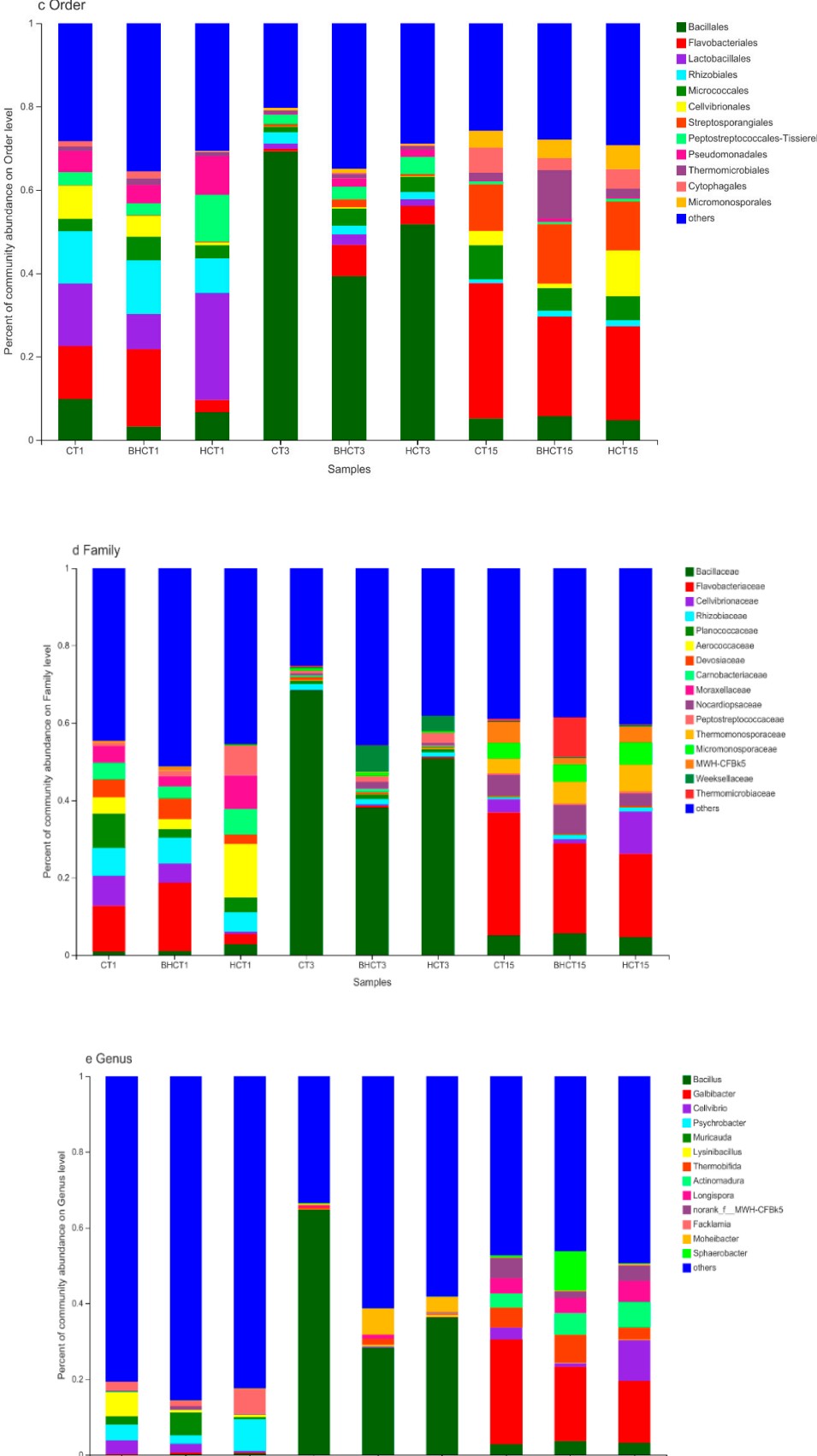

**Figure 2.** *Cont.*

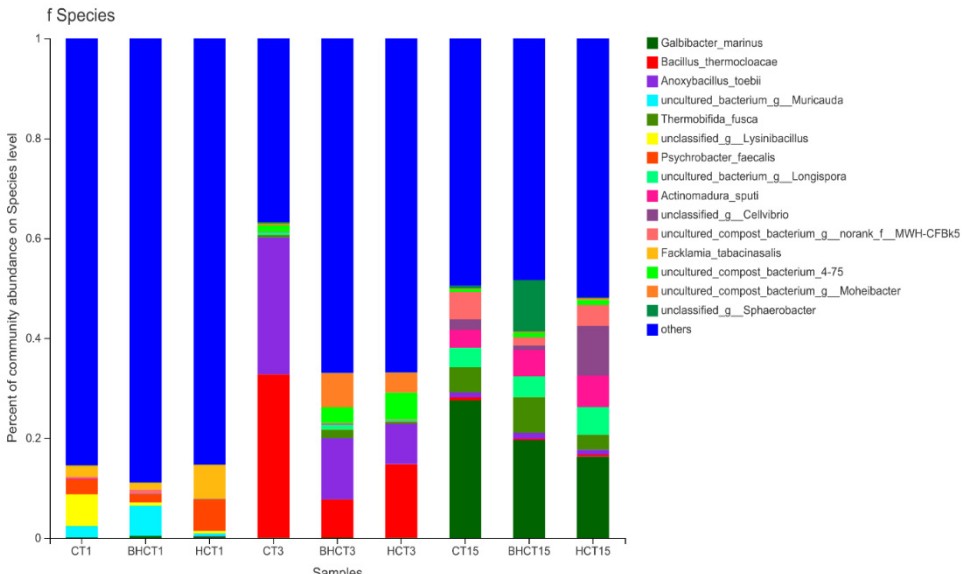

**Figure 2.** The distribution histogram of the relative abundance of heavy-metal-resistant bacteria from the phylum to species level.

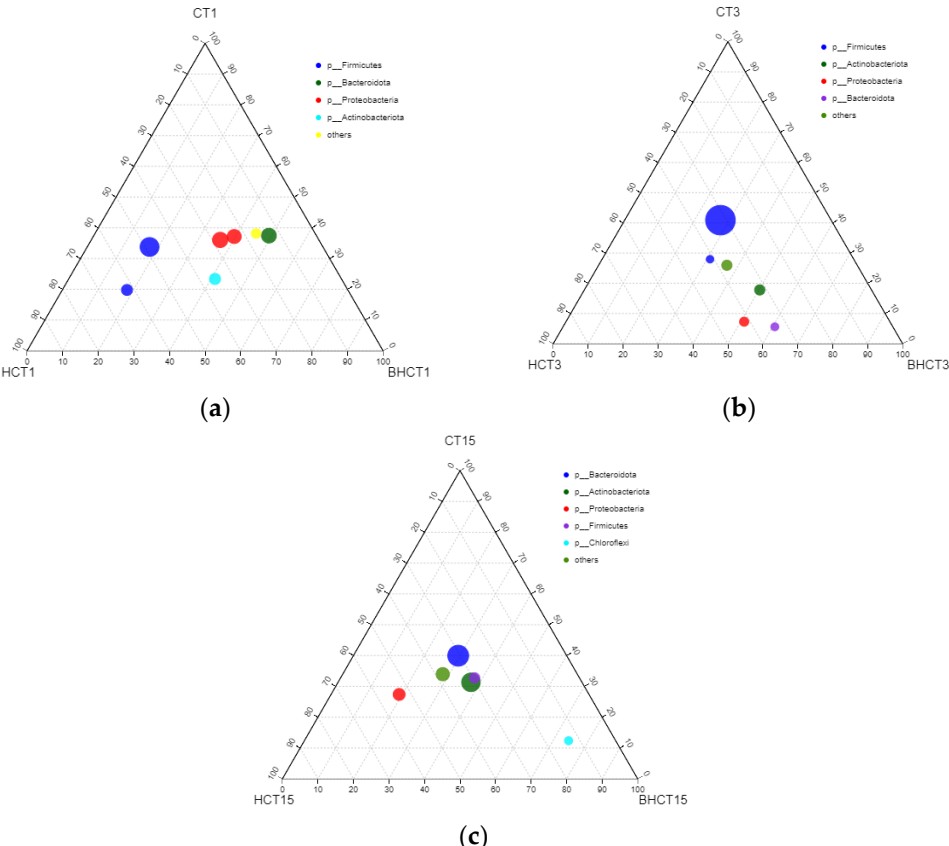

**Figure 3.** Ternary diagram at the phyla of the bacteria community among CT, BHCT, and HCT during three different composting stages: (**a**) Day 1, (**b**) Day 3, (**c**) Day 15.

As previous reported, Firmicutes and Actinobacteria largely caused the HMRB to disseminate into the environment, and Actinobacteria was the predominant HMRB phyla [28,29]. Awasthi et al. [11] pointed out that Firmicutes, Proteobacteria, Bacteroidota, Actinobacteriota, and Chloroflexi were the most dominant HMRB-source phyla. In the present study, the reduction of identified Firmicutes might be explained by the removal of HMRB un-

der biochar and hyper-thermal inoculum amendments. The elevated relative abundance of Proteobacteria, Bacteroidota, Actinobacteriota, and Chloroflexi probably contributed to the action of biochar and hyper-thermal inoculum and effectively altered the habitat condition of the heavy metals and nutrients. Awasthi et al. [4] found that the relative abundance of bacterial communities changed with an increased biochar concentration, which suggests that biochar amendment can regulate potential bacterial hosts by changing the HMRB. Sui et al. [30] illustrated that the HMRB distribution is probably caused by bacterial communities altered by organic matter. The prominent characteristics of microbial community dynamics in hyperthermophilic composting, e.g., in Thermoactinomycet-aceae, were determined by the hyperthermophilic phase, changing the structure of dissolved organic matter, which can mediate the passivation of HMs [14,31]. This illustrates that the HMRB varied according to environmental conditions. Biochar and hyper-thermal inoculum amendments can regulate main bacterial communities to modify heavy metals by controlling modulations of the metabolic environment of microorganisms [14,28].

*3.2. Diversity Analysis of HMRB*

The distribution of HMRB compositions and Fisher's exact test bar plot on the genus level among all treatments is presented in Figure 4a–f. The addition of biochar and hyper-thermal inoculum exhibited a significant influence on the variation of HMRB. As shown in Figure 4, the bacterial community was dominated by the genera *Bacillus*, *Galbibacter*, *Cellvibrio*, *Thermobifida*, and *Actinomadura*. The results revealed that the abundance of *Bacillus* in BHCT3 accounted for 20% of the total abundance, which was lower than that in the CT3 treatment (46%). In CT15, BHCT15, and HCT15 in the maturation phase, *Galbibacter* (43%, 30%, and 25%), *Cellvibrio* (15%, 4.5%, and 50%), *Thermobifida* (28%, 40%, and 17%) and *Actinomadura* (23%, 34%, and 41%) became the dominant bacterial communities among all compost samples. The abundance of bacterial communities in each treatment was compared two-to-two by the Fisher's test to verify the difference. These results showed that the abundance of ubiquitous bacterial communities at the genus level in different treatments revealed significant differences ($p < 0.05$). In particular, *Galbibacter*, *Thermobifida*, *Sphaerobacter*, and *Actinomadura* were dramatically different in CT15 and BHCT15 ($p < 0.05$). In the same phase, the abundance of bacteria in the BHCT treatment was always more abundant than that of the CT treatment. Moreover, *Bacillus*, *Moheibacter*, and *Ammoniibacillus* in CT3 and BHCT3 as well as *Confluentibacter*, *Muricauda*, *Lysinibacillus*, *Psychrobacter*, and *Corynebacterium* in CT1 and BHCT1 revealed significant differences ($p < 0.05$).

Overall, these dominant bacteria were considerably responsive to the regulation of HMRB and the abundance altered by the stress of biochar and hyper-thermal inoculum [31,32]. The fertilization of the compost significantly increased *Galbibacter* relative to most chemical characteristics, reducing the *Mizugakiibacter*, which was highly sensitive to Hg, Pb, Zn, and Cd [33,34]. Arsenic (As) methylation during manure composting was primary attributed to *Sphaerobacter* (Chloroflexi), which are more active than other microbes [35]. Co-occurrence patterns revealed that *Actinomadura* is a significant heavy-metal-resistant gene subtype in soil contaminated with nickel iron, arsenic, copper, and zinc, and the nickel phytoextraction promoted the *Psychrobacter* bacterium [36,37]. The bacterial community profiles of *Thermobifida*, *Bacillus*, and *Corynebacterium* as potential HMRB hosts such as Cu, Zn, Cd, and Pb varied with the addition of biochar and thermo-tolerant bacterial inoculation [4,38]. The hyper-thermal inoculum generated distinctive bacterial communities. The predominant community was the thermophilic genus, which can produce abundant enzymes, such as *Actinomadura*, which is a heavy-metal-resistant gene subtypes and can produce highly thermostable xylanase [14]. The application of biochar altered the bacterial community and influenced heavy metal stress and microbial dynamics during composting, such as the dominant genus *Moheibacter* [39]. It is suggested that amendments of biochar and hyper-thermal inoculum probably change the metabolic environment and alter the formation and abundance of HMRB hosts, as biochar and hyper-

thermal inoculum most likely regulates bacterial communities by contributing a favorable habitat for microbial metabolism [4,38].

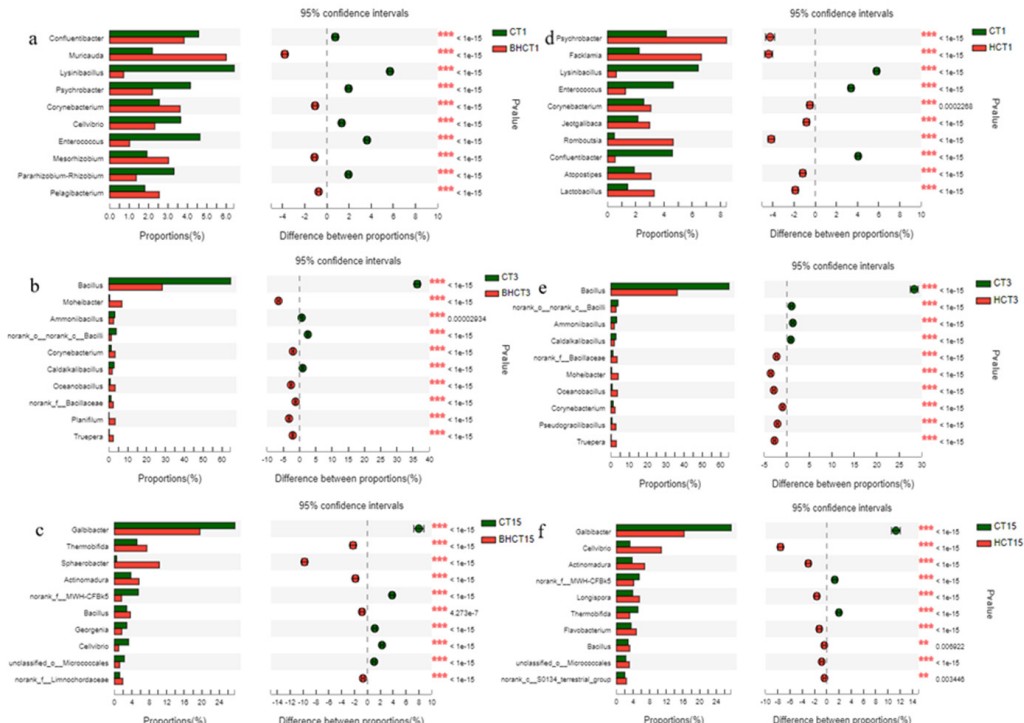

**Figure 4.** Paired Fisher's exact test bar plot on the genus level between BHCT, HCT, and CT during three different composting stages: (**a**) CT1-HCT1, (**b**) CT3-HCT3, (**c**) CT15-HCT15, (**d**) CT1-BHCT1, (**e**) CT3-BHCT3 (**f**) CT15-BHCT15.

### 3.3. Correlation Analysis among HMRB Communities and Selected Factors

3.3.1. Correlation between the HMRB and Physicochemical Parameters

In Table 2, the physicochemical parameters of CT, HCT, and BHCT in the initial and final stages are shown. Correspondingly, redundancy analysis (RDA) was used to identify the differences among the bacterial community and environmental variables (pH, EC, temperature, $NH_4^+$-N, $NO_3^-$-N, TN, C/N, MC, and GI at the genus level (Figure 5). The first and second axes explained 66.72% and 21.39% of the variation in the species, respectively. The selected factors and parameters among potential host bacteria showed a significant correlation ($p < 0.05$) with bacterial community. As shown in Figure 5, the temperature showed a significant correlation with CT3, BHCT3, and HCT3, and the order of influence is CT3 > HCT3 > BHCT3. The parameters of TN, $NO_3^-$-N, and *Bacillus* were closely related to the temperature in the Day 3 group. Furthermore, TN, $NO_3^-$-N, and GI showed significant correlations with CT15, BHCT15, and HCT15, and the order is CT3 > HCT3 ≈ BHCT3, while *Galbibacter*, *Thermobifida*, and *Flavobacterium* were remarkably related to TN, $NO_3^-$-N, and GI within the maturation group. However, *Confluentibacter*, *Psychrobacter*, *Lysinibacillus*, and *Muricauda* showed a significant correlation with C/N and MC in the Day 1 group.

**Table 2.** The main physicochemical properties of CT, HCT, and BHCT in the initial and final stages.

| Parameter | CT (Initial Value) | BHCT (Initial Value) | HCT (Initial Value) | CT (Final Value) | BHCT (Final Value) | HCT (Final Value) |
|---|---|---|---|---|---|---|
| pH | $6.79 \pm 0.07$ | $6.83 \pm 0.05$ | $7.23 \pm 0.03$ | $8.15 \pm 0.06$ | $8.02 \pm 0.11$ | $7.92 \pm 0.06$ |
| TN | $1.74 \pm 0.02$ | $1.65 \pm 0.07$ | $1.80 \pm 0.09$ | $0.95 \pm 0.03$ | $1.19 \pm 0.05$ | $0.98 \pm 0.01$ |
| C/N | $30.03 \pm 0.76$ | $32.41 \pm 0.84$ | $35.77 \pm 0.93$ | $24.37\pm$ | $20.73\pm$ | $21.41\pm$ |
| As (mg kg $^{-1}$) | $6.07 \pm 0.04$ | $6.68 \pm 0.01$ | $6.37 \pm 0.01$ | $4.33 \pm 0.03$ | $2.49 \pm 0.04$ | $3.61 \pm 0.01$ |
| A-As (mg kg $^{-1}$) | 0.02 | 0.02 | 0.02 | 0.01 | 0.01 | 0.01 |
| Pb (mg kg $^{-1}$) | $22.25 \pm 0.05$ | $21.92 \pm 0.07$ | $20.25 \pm 0.05$ | $21.03 \pm 0.04$ | $19.07 \pm 0.02$ | $20.11 \pm 0.07$ |
| A-Pb (mg kg $^{-1}$) | $3.45 \pm 0.01$ | $3.35 \pm 0.01$ | $3.35 \pm 0.02$ | $1.25 \pm 0.02$ | $1.06 \pm 0.03$ | $1.27 \pm 0.01$ |
| Cd (mg kg $^{-1}$) | $0.48 \pm 0.01$ | $0.48 \pm 0.02$ | $0.43 \pm 0.01$ | $0.40 \pm 0.01$ | $0.35 \pm 0.01$ | $0.44 \pm 0.03$ |
| A-Cd (mg kg $^{-1}$) | 0.14 | 0.15 | 0.13 | 0.10 | 0.08 | 0.09 |
| Cr (mg kg $^{-1}$) | $38.17 \pm 0.27$ | $37.11 \pm 0.34$ | $37.95 \pm 0.25$ | $51.41 \pm 0.16$ | $47.37 \pm 0.1$ | $52.69 \pm 0.15$ |
| A-Cr (mg kg $^{-1}$) | $5.11 \pm 0.05$ | $5.41 \pm 0.07$ | $5.57 \pm 0.03$ | $4.19 \pm 0.06$ | $3.30 \pm 0.05$ | $3.98 \pm 0.08$ |

Note: Values indicate mean $\pm$ standard deviation based on the samples with 3 replications. TN: total nitrogen; C/N ratio: carbon-to-nitrogen ratio. A-P: available phosphorus; Cr: chromium; A-Cr: available chromium; Pb: lead; A-Pb: available lead; Cd: cadmium; A-Cd: available cadmium; As: arsenic; A-As: available arsenic.

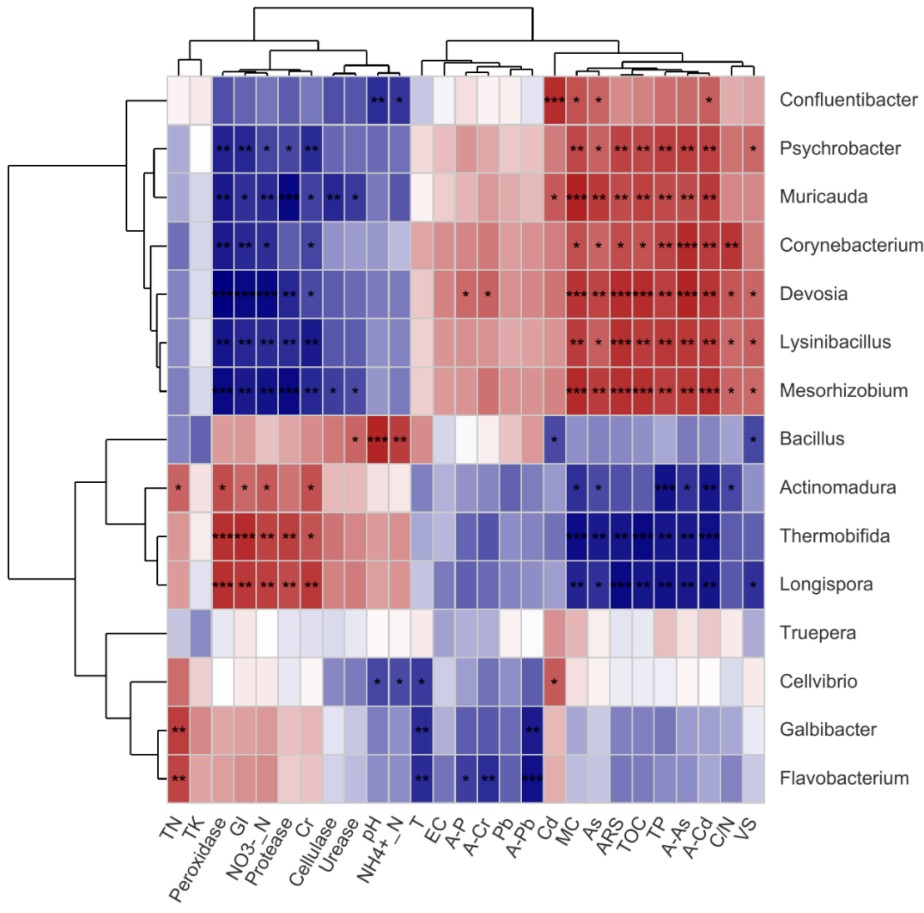

**Figure 5.** Heat map of the 15 most abundant genera showing correlations between HMRB bacterial community and environmental factors, enzyme activity, and heavy metals. Vertical: bacterial information; horizontal: selected factors. The left and acroscopic cluster tree reflect species and factors, respectively. Different colors indicate different correlation coefficients (r). "*" $p < 0.05$, "**" $p < 0.01$, and "***" $p < 0.001$. T: temperature; EC: electrical conductivity; GI: germination index; TOC: total organic carbon; TN: total nitrogen; C/N ratio: carbon-to-nitrogen ratio. TK: total potassium; TP: phosphorus; A-P: available phosphorus; Cr: chromium; A-Cr: available chromium; Pb: lead; A-Pb: available lead; Cd: cadmium; A-Cd: available cadmium; As: arsenic; A-As: available arsenic; ARS: arylsulfatase.

Combined with Figure 6, parameters among the environmental variables (e.g., pH, temperature, HM, and water-soluble carbon and nitrogen) and HMRB potential host bacteria registered significant correlations with bacterial community. *Bacillus* as a potential HMRB host was the dominant genus and was positively correlated with temperature during the composting process [4], and pH was positively related to *Bacillus* and *Galbibacter* [33]. Microbial inoculation enriched the abundance of the genera *Thermobifida*, *Bacillus*, and *Flavobacterium*, which served as or were related to potential HMRB hosts, enhancing the organic humification and ammonification [12]. Simultaneously, the bacterial community was primarily impacted by environmental elements. The close connection transformation might be attributed to the complex pathways of metabolic functions, which are influenced by cellular processes, genetic information, and the related environment condition [13]. In BHCT and HCT samples, biochar and hyper-thermal inoculum amendments can better resist excessively stressed environmental parameters. Overall, the exogenous additive of biochar and hyper-thermal inoculum affect the HMRB profile primarily due to self-adapting micro-environmental factors that indirectly affect the dominant functional bacterial community [4].

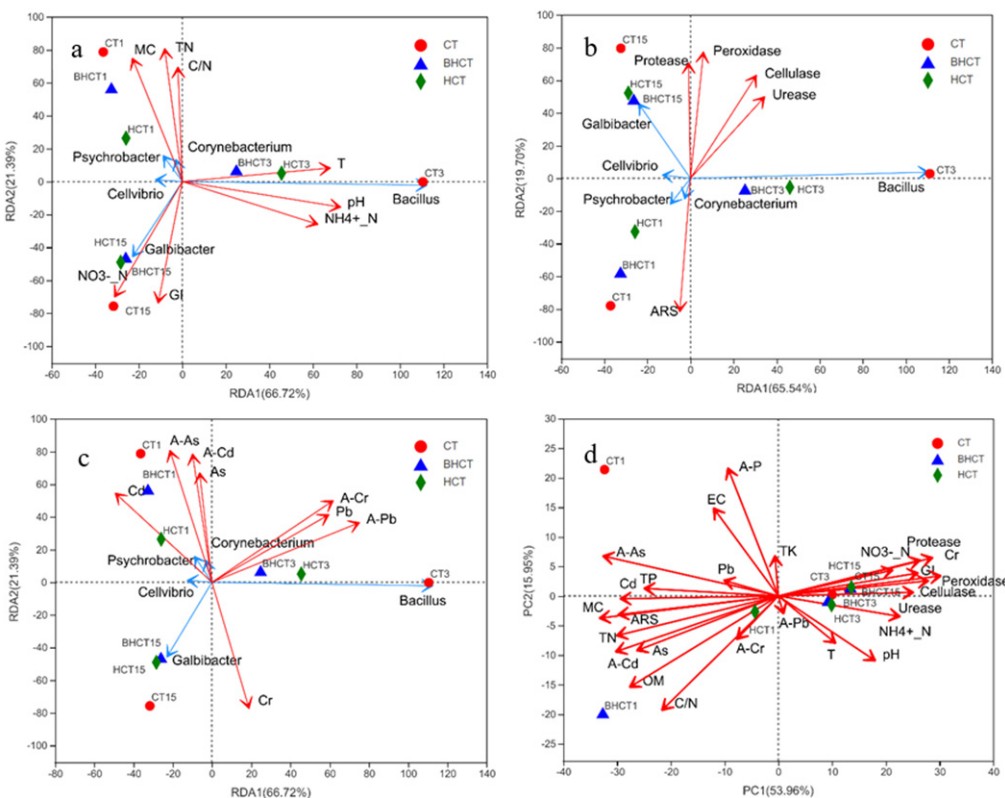

**Figure 6.** Redundancy analysis of HMRB bacterial community in relation to (**a**) environmental factors, (**b**) enzyme activity, and (**c**) heavy metals. (**d**) Principal coordinates analysis.

### 3.3.2. Correlation between the HMRB and Enzyme Activity

As shown in Figure 5, the relationship between HMRB and enzymes varied with biochar and hyper-thermal inoculum additions during the composting process. The distribution of enzymes was relatively concentrated and presented a strong correlation with the composting process stage, while the distribution without the addition of biochar or hyper-thermal inoculum was widely scattered. In addition, *Bacillus* shows a close relationship with cellulase and urease, and the relationship was obviously positive in the Day 3 group. *Galbibacter* is highly associated with peroxidase and protease, and this relationship was obviously positive in the Day 15 group. In contrast, *Corynebacterium* and *Psychrobacter* showed clear increasing trends with ARS in the Day 1 group.

It is well known that microbe inoculants can stimulate microbial activity to promote the composting process. The addition of biochar significantly decreased the heavy metal concentrations, increased the diversity of HM-resistant bacteria, and resulted in more synthesis and secretions of microbial enzymes due to increases in heavy-metal-resistant bacterial communities [13,40]. Zhu et al. [41] pointed out that applied biochar might improve hydrolytic enzyme generation by the increase of *Actinobacteria*. Biochar addition increases the abundance of *Bacillus* as HM-resistant bacteria and promotes the release of cellulase for organic matter decomposition, contributing to HM passivation, consistent with [42,43]. Harindintwali et al. [44] reported that the addition of microbial inoculants can generate various desired enzymes. Yu et al. [14] also pointed out that a thermus genus can produce hydrolytic enzymes and catalases as the predominant community during the hyperthermophilic composting. The application of biochar and hyper-thermal inoculum enhanced the activities of most functional enzymes, which benefits the evolution of functional enzymes in composting and weakens the relationship between enzyme activities and different treatments, hence strengthening the ability to resist heavy metal stress [4]. *Bacillus*, *Galbibacter*, *Corynebacterium*, and *Psychrobacter* are potential hosts for HMRB [4,33,36]. The correlations indicate that the application of biochar and hyper-thermal inoculum acting as an activator strongly regulates HMRB composition, ultimately improving enzyme activities for degradation process optimization.

### 3.3.3. Correlation between HMRB Communities and Heavy Metals

In Table 2, the heavy metal parameters of CT, HCT, and BHCT in the initial and final stages are shown. During the composting, heavy metals interacted with HMRB communities, revealing the effects of biochar and hyper-thermal inoculum application on the HMRB structure. The correlation between heavy metals and HMRB composition was studied via RDA. According to Figure 5, the sample points show a prominent dispersity. Those of BHCT and HCT are relatively concentrated, while those of CT are relatively dispersed. In addition, the *Bacillus* was positively related to Pb, A-Pb, and A-Cr. The *Galbibacter* genus showed a positive correlation with Cr, while the *Cellvibrio*, *Corynebacterium*, and *Psychrobacter* showed a positive correlation with A-As, As, A-Cd, and Cd. As previously reported, these dominant bacteria were usually considered major contributors to the bio-removal/persistence of HMRB, which can resist various extreme environments [45]. It is indicated that biochar and hyper-thermal inoculum amendments may weaken HM stress, which can induce the HM resistance gene and decrease the correlation with HMRB [46].

This is also consistent with the results of Figure 6. The high temperatures of BHCT and HCT were positively correlated with pH, but negatively correlated with the major HMRB distribution in the Day 3 group, suggesting that a higher pH could improve the passivation effect of HMs. It can be inferred that there are abundant HMRB and HT-resistant bacteria [47,48]. The hyper-thermal inoculum can increase the temperature and maintain a high degradation ability of organic matter, thereby enhancing the humic substances, the main driving parameter affecting the bioavailiblity of HMs [48,49]. Simultaneously, Figure 5 indicates a strong negative correlation between enzyme activities and heavy metals (except ARS), which further suggests that biochar and hyper-thermal inoculum amendments might activate enzyme activities by reducing the heavy metal values and accelerating compost maturity, while a higher humification degree can strengthen the passivation of HMs, similar to previous research [50]. The results indicate that biochar and hyper-thermal inoculum amendments altered HM toxicity and physicochemical properties, thereby changing the superior HMRB bacterial fate by regulating the potential bacterial hosts [51].

## 4. Conclusions

Biochar and hyper-thermal inoculum addition can regulate the abundance of HMRB under HM stress, enhancing bacterial reproduction. The prominent characteristics of microbial community dynamics in hyperthermophilic composting change the structure of dissolved organic matter, which can mediate the passivation of HMs. The bacterial

community profiles of *Thermobifida*, *Bacillus*, and *Corynebacterium* as potential HMRB hosts, such as Cu, Zn, Cd, and Pb, varied with the addition of biochar and thermotolerant bacterial inoculation. *Bacillus*, *Galbibacter*, *Corynebacterium*, and *Psychrobacter* as HMRB potential hosts, the correlations of which indicated that the application of biochar and hyper-thermal inoculum acting as an activator strongly regulated the HMRB composition, ultimately improved the enzyme activities in optimizing the degradation process. The strong negative correlation of enzyme activities with heavy metals, except ARS, further suggests that biochar and hyper-thermal inoculum amendments might activate enzyme activities by reducing heavy metal values. In addition, the dominant phyla of Proteobacteria, Bacteroidota, Actinobacteriota, and Chloroflexi were enriched, but this was not the case with Firmicutes. The genus of *Galbibacter*, *Thermobifida*, *Sphaerobacter*, and *Actinomadura* were dramatically different in CT15 and BHCT15. Therefore, the study can help to better understand the successions of HMRB under the biostimulation of compost amended with biochar and hyper-thermal inoculum.

**Author Contributions:** Q.Z., T.Z., Q.X. and N.A. analyzed and interpreted the data regarding the co-composting. Q.Z. performed the enrichment and adaptation of hyper-thermal inoculum and was a major contributor in writing the manuscript. All authors have read and agreed to the published version of the manuscript.

**Funding:** This research was funded by National Key Research and Development Program of China (No. 2020YFC1806402); and Major key and core technology research project, Science and Technology Plan of Shenyang in 2020 (No. 20-202-4-37).

**Institutional Review Board Statement:** Not applicable.

**Informed Consent Statement:** Not applicable.

**Data Availability Statement:** All data generated or analyzed during this study are included in this published article.

**Conflicts of Interest:** The authors declare that there is no conflict of interest.

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
