# Peer review of "The Addition of Biochar and Hyper-Thermal Inoculum Can Regulate the Fate of Heavy Metals Resistant Bacterial Communities during the Livestock Manure Composting"

_fermentation, doi:10.3390/fermentation8050207_

Round 1
Reviewer 1 Report
The authors reported addition of biochar and hyper-thermal inoculum can regulate the fate of heavy metals resistant bacterial communities during the livestock manure composting. This work may be acceptable for publication after addressing the comments below and revising the manuscript.
- The authors need to improve a few errors (e.g., reference section; abbreviation).
- How to prepare the biochar? Please add the process of biochar preparation.
- The authors should state the more detailed phenomena by biochar and hyper-thermal inoculum.
- Please revise the legends in all Figures. Can not see.
- What is the final values (e.g., pH, N, HMs…)
- Which software did you use for PDA…?
- Please improve the conclusion section.
- Please check the HMs in initial mixture.
- In my opinion, it is better to add the schematic diagram for readers.
Reviewer 2 Report
Figures: need better resolution from figures 2 until to 6; similar table 1 is difficult for reader - please try to change if possible.
Reviewer 3 Report
Dear Authors,
The article: Addition of biochar and hyper-thermal inoculum can regulate the fate of heavy metals resistant bacterial communities during the livestock manure composting, is extremely interesting. The research carried out by the authors is detailed and provides valuable information on the communities of microorganisms and their interrelationships depending on the changing composting additives.
The submitted manuscript presents an added value to the scientific area of ​​compost processing and increases its application potential. In general, I can recommend considering this manuscript for publication after obtaining clarifications from the Authors on the issues given below:
General comment:
With such extensive and detailed research, the conclusions are quite "sparing", please elaborate further on this part, the results already contain many statements that can also be included in the conclusions. Fig. 4 is very interesting but in its present form completely illegible - please correct it. The order of Figs. 5 and 6, and the work is Fig. 6 first and then 5, I think is confusing and misleading to the reader.
I hope that my suggestions will help to improve the paper and its readability before it can be published in Fermentation.
